# Displacement Detection Decoupling in Counter-Propagating Dual-Beams Optical Tweezers with Large-Sized Particle

**DOI:** 10.3390/s20174916

**Published:** 2020-08-31

**Authors:** Xunmin Zhu, Nan Li, Jianyu Yang, Xingfan Chen, Huizhu Hu

**Affiliations:** 1State Key Laboratory of Modern Optical Instrumentation, College of Optical Science and Engineering, Zhejiang University, Hangzhou 310027, China; 11630014@zju.edu.cn (X.Z.); 3160101439@zju.edu.cn (J.Y.); mycotty@zju.edu.cn (X.C.); huhuizhu2000@zju.edu.cn (H.H.); 2Quantum Sensing Center, Zhejiang Lab, Hangzhou 310000, China

**Keywords:** optical tweezers, optical trap, acceleration, decoupling

## Abstract

As a kind of ultra-sensitive acceleration sensing platform, optical tweezers show a minimum measurable value inversely proportional to the square of the diameter of the levitated spherical particle. However, with increasing diameter, the coupling of the displacement measurement between the axes becomes noticeable. This paper analyzes the source of coupling in a forward-scattering far-field detection regime and proposes a novel method of suppression. We theoretically and experimentally demonstrated that when three variable irises are added into the detection optics without changing other parts of optical structures, the decoupling of triaxial displacement signals mixed with each other show significant improvement. A coupling detection ratio reduction of 49.1 dB and 22.9 dB was realized in radial and axial directions, respectively, which is principally in accord with the simulations. This low-cost and robust approach makes it possible to accurately measure three-dimensional mechanical quantities simultaneously and may be helpful to actively cool the particle motion in optical tweezers even to the quantum ground state in the future.

## 1. Introduction

Optical tweezers (OT), as an intriguing tool in various areas such as cell biology, weak mechanics sensing and quantum physics, enjoys increasingly attractive prospects [1,2,3]. A particle levitated in OT is isolated from the thermal noise of clamping, which is a fundamental, unavoidable source of dissipation in a traditional mechanical oscillator [4,5]. Moreover, the optical interference method can be easily used in OT to measure displacement with excellent spatial and temporal resolution. Therefore, OT in high-vacuum can measure ultra-weak acceleration up to the nano-g scale, as the state of art level in a mechanical sensing application [4]. There are many proposals and experiments in exploring fields including non-Newtonian gravitation at sub-millimeter length scales, as well as the precise measurement of static characteristics and temporal variations of earth gravity [6,7].

The minimum measurable force in a measurement of bandwidth b is $F_{\min}=\sqrt{4k_{B}Tm\Gamma_{M}b}$ for a microsphere with mass m, radius r and density ρ when laser cooling not exerted, where viscous damping factor $\Gamma_{M}=16P/\pi \rho \nu r$ [8]. $k_{B}$ is Boltzmann constant and T is temperature of the sphere surface. P is background gas pressure and ν represents the mean speed of gas molecules. Then the minimum measurable acceleration $a_{\min}=\frac{F_{\min}}{m}=\frac{4}{\rho \pi }\sqrt{\frac{3k_{B}TPb}{\nu }}\frac{1}{r^{2}}\propto \frac{1}{r^{2}}$ since other variables in the equation are all completely independent of r. Thus, weaker acceleration can be detected when larger particles suspended. At present, there are two OT schemes for levitating large-sized particles: a single vertically upward beam or horizontal counter-propagating dual-beams. In the former scheme, axial radiation pressure is balanced with gravity on the particle, while pressures of the two beams cancel out in the latter scheme. The maximum diameter of particle currently levitated in the two solutions is 14 μm [9] and 10 μm [10], respectively. However, the latter usually has a larger response bandwidth, due to radial radiation pressure being stronger than axial one in the same beam. Moreover, it has an applicability of working in a microgravity environment.

Owing to its axial symmetry, the particle centroid is naturally on the optical axis in OT with a single beam and a spherical particle, but this is not truly a misaligned OT with multi-beams. Beam misalignment is defined as the incomplete coincidence of the focuses and optical axes of the multi-beams in OT. The distance between the particle centroid and the optical axis $x_{x}$, are usually positively correlated with, the radial distance of focuses of two beams. As a special case, the former one is the half of the latter one if the structure and intensity of two beams are identical. Coupling becomes notable as the sphere diameter is close to the order of the beam waist diameter at focus and the particle centroid deviates from the optical axis. However, the relation between coupling-detection and the ratio of sphere diameter and beam waist at focus has not been quantitatively described [11]. To a certain extent, it limits acceleration detection performance. Unfortunately, it is challenging to ensure the coaxiality of counter-propagating beams. Aiming focuses of two beams at the same pinhole is commonly used for alignment in OT. However, Distance $x_{x}$ of only about one micrometer can be reached due to restriction by the coaxiality of the tubular pinhole and the two beams. This error is defined as the distance between the particle centroid and the optical axis when forces on it are balanced. Increasing beam waist can weaken coupling, but it also drastically reduces detection sensitivity. Furthermore, detection, already restricted by other noises such as Johnson noise, deteriorates seriously. Response bandwidth also decreases and the chance of applying OT in realms of high-speed particle motion is missed.

Although a spherical particle of three micrometers in diameter [12,13] or 10 micrometers [10] has been levitated in experiments in OT of dual-beams, the maximum diameter according to theoretical research on particle translation detection regime is only one micrometer [11]. We chose to suspend a ball of 10 μm in diameter both in the simulation and the experiment in OT of dual-beams. Moreover, its coupling was analyzed with forward-scattering far-field images acquired by the Fresnel diffraction method. There is a novel method proposed to suppress coupling, adding a variable iris in front of the photodetector on each axis. Experiments show a reduction of 49.1 dB in coupling-detection ratio radially and that of 22.9 dB axially, which basically corresponds with simulations. These make it possible to accurately measure three-dimensional mechanical quantities with OT simultaneously and may contribute to do further operations beyond it such as actively cooling the particle to quantum ground state in low gas pressure in basic physics research. Moreover, the original optical structures of levitation and detection do not need to be changed, because these three variable irises are set just behind all optical components in OT. A laser beam profiler is also employed for the temporary recording of far-field images. Importantly, the above methods have advantages of low cost and good structural compatibility.

## 2. Theory

### Displacement Detection and Coupling

Most mechanical quantities to be measured in OT are directly related to the levitated particle centroid displacement detection. The displacement detection methods mainly fall into two camps: recording far-field interference images directly [14,15] with sensors such as CCD or measuring the intensity of each part of the far field interference image with sensors such as four-quadrant photodetectors (QPD) and a balanced photodetector (BD) [11,12,16,17]. Although the former can obtain more information from the image, its detection sensitivity is impaired by a time-consuming image transmission and process. Thus, the latter one should only be used when pursuing a detection of ultra-low mechanical quantities.

The typical QPD scheme of particle centroid displacement detection in OT is shown in Figure 1. If the optical radiation force for levitating particles is provided by beam No. 1 only, it is then called single beam OT. In contrast, sometimes there is a counter-propagating beam No.2. This is reflected by BS and focused by a condenser, thus forming counter-propagating dual-beam OT.

As shown in Figure 1, one radial direction of beam is defined as the x-axis, and the beam propagates along the positive z direction. This set of coordinates is always used below. Assuming that the voltages obtained by converting light intensity on the four photosensitive surfaces are $V_{k},\;\mathrm{values}\;\mathrm{for}\;k=1\;to\;4$, respectively. Supposing $V_{T}=V_{1}+V_{2}+V_{3}+V_{4}$, the final detection response of each axis will be [13]:(1)$$\left\{ \begin{array}{l} V_{x}=\frac{V_{1}-V_{2}-V_{3}+V_{4}}{V_{T}}<V_{T}>\\ V_{y}=\frac{V_{1}+V_{2}-V_{3}-V_{4}}{V_{T}}<V_{T}>\\ V_{z}=V_{T}-<V_{T}> \end{array}\right.,$$
where symbol <> means time-domain average. The output voltage differences of two half photosensitive surfaces in QPD $V_{1}-V_{2}-V_{3}+V_{4}$ and $V_{1}+V_{2}-V_{3}-V_{4}$, will vary with laser intensity fluctuation even when the levitated particle remains stationary in the position, deviating from the laser optical axis. Hence, the above differences are divided by $V_{T}$ to suppress distortion coming from laser intensity fluctuation. Then we use the time-average $<V_{T}>$ to keep the unit of $V_{x}$ and $V_{y}$ being voltage for more convenient analysis of detection. We subtract $V_{T}$ with $<V_{T}>$ to acquire relatively smaller AC component and subsequent electronic signal amplification of enhancing the detection sensitivity in z direction. The response of BD is the same as of QPD, while smaller photosensitive surfaces are used for detection at higher bandwidths and of lower electricity noises. For convenience, only the QPD method will be discussed below.

Assuming that $V_{ij}$ is *i*-axis detection response resulting from the *j*-axis motion, i=x, y, z and j=x, y, z are always applicable hereinafter if not mentioned. The above total response of *i*-axis detector can be expressed as:(2)$$V_{i}=\sum_{j=x,y,z}V_{ij}.$$
The levitated particle in OT is always collided by the surrounding gas molecules, thereby generating irregular thermal motion. It is a kind of main noise source of displacement detection under relatively high gas pressure. The thermal motions of the particles on three-axis are all random and uncorrelated to each other. The motion information of other axes will become noise in the displacement detection signal of one specific axis. The definition of i-axis coupling-detection ratio resulting from the j-axis motion will then be:(3)$$R_{ij}=\frac{\langle {V_{ij}}^{2}\rangle }{\langle {V_{ii}}^{2}\rangle },\;i\neq j.$$

The logarithmic form $R_{ij}\left(\mathrm{dB}\right)=10\log_{10}R_{ij}$ is often used there. There are six items for tri-axial displacement coupling-detection, but only $R_{\mathrm{xy}}$, $R_{\mathrm{xz}}$ and $R_{\mathrm{zx}}$ need to be considered owing to axial symmetry. In general, the motion range of the particle centroid does not exceed the micrometer scale and is much smaller than the beam waist of several micrometers. Therefore, the detection response is nearly linear with displacement. The *j*-axis motion detection sensitivity of *i*-axis detector is defined as:(4)$$\beta_{ij}=\frac{\partial V_{ij}}{\partial x_{j}},$$
where $x_{j}$ is the particle displacement of *j*-axis. The particle is on the optical axis and is 10 μm away from the focus of Beam No.1, along the beam propagation direction when $x_{x}=x_{y}=x_{z}=0.$

Assuming the detection response $dV_{ij}=\beta_{ij}dx_{j}+\gamma_{ij}{\left(dx_{j}\right)}^{2}+O\left({\left(dx_{j}\right)}^{2}\right)$ when the particle moves to a specific position $x_{j}$. Symbol d represents differential and $O\left({\left(dx_{j}\right)}^{2}\right)$ means higher-order infinitesimals of ${\left(dx_{j}\right)}^{2}$. Then the second-order nonlinearity of the displacement measurement will be:(5)$$\chi_{ij}=20\log_{10}(\left|\frac{\gamma_{ij}\sqrt{<{x_{j}}^{2}>}}{\beta_{ij}}\right|).$$

The random collision noise of gas molecules determines the standard deviation of j-axis displacement $\sqrt{{\langle x_{j}\rangle }^{2}}=\sqrt{k_{B}T/k_{j}}$ in the absence of laser cooling [11]. $k_{B}$ is the Boltzmann constant and T is the temperature of the sphere surface. $k_{j}={\left(2\pi f_{0,j}\right)}^{2}m$ is the coefficient of elasticity of radiation force of j-axis. $f_{0,j}$ is the frequency of resonant peak in displacement power spectrum and m is the mass. Taking the z-axis experimental data in the following Section 5.2 as an example when T=300 K, $\sqrt{{\langle x_{z}\rangle }^{2}}\approx 8.3\;\mathrm{nm}$. $V_{\mathrm{zj}}$ changes slowly on the scale of laser beam waist level, about several micrometer. Thus, the nonlinearity of displacement measurement in OT with QPD method is intuitively small.

## 3. Materials and Methods

### 3.1. Setup in Simulations and Experiments

We used the same structure explained in Figure 1 with horizontal dual-beams in simulations and experiments. The silica sphere (Model 904368-2G, Sigma-Aldrich, Inc., St. Louis, MI, USA) diameter was 10 μm. Other parameters were as follows: The objective and condenser were lens of the same model (A280-C, Thorlabs, Inc., Newton, NJ, USA) and their numeric aperture (NA) were both 0.15; laser (Model Opus 1064, Laser Quantum, Inc., Stockport, UK) wavelength in vacuum $\lambda_{0}$ was 1.064 μm; the ambient medium was air, thus its refractive index $n_{\mathrm{med}}=1$. Consequently, the diameter of incident beam waist at focus was $2\lambda_{0}{n_{\mathrm{med}}}^{2}/\left(\pi \mathrm{NA}\right)\approx$4.5 μm. The focal length of condenser $f_{1}$ and objective $f_{2}$ were both 18.4 mm. The QPD (Model KY-SQP-7, Keyang Photonics, Inc., Beijing, China) was placed 0.2 m behind the condenser. The intensity of each beam was 800 mW. The objective and condenser were both mounted on adjustment frames (Model POLARIS-K05F6, Thorlabs, Inc.) that could move along the z-axis. First, we checked and ensured that the focal points of the two lensed coincide by shearing interferometer. Second, the focused of two beams were set to 20 μm away by carefully turning the screws of 130 threads per inch (TPI) on the adjustment frames, about 37 degree. Moreover, finally, the spherical particle was deduced to be levitated at the center of the two foci since the structure and intensity of two beams were identical. Thus, the particle was nearly on the optical axis and was about 10 μm away from the focus, along the beam-propagation direction.

### 3.2. Forward Scattered Far Field Computation

#### 3.2.1. Computation Principle

Increasing detection information is a feasible way to coupling suppression. The QPD method can only obtain the total intensities of four parts of the far-field interference image, while image recording methods are defective in accuracy by getting intensities of many points. Herein we try to find out the characteristics of a far-field interference image, which are more closely related to the positional change of a particle by a combination of simulation of and experiment on the QPD detection regime.

Triaxial QPD detection responses of OT with single Gaussian beam structure were calculated by means of Rayleigh scattering [16,17], Mie scattering [18] and the extended boundary condition method [11]. Only the last method can analyze large-sized particles up to several micrometers with acceptable time–space complexity of computation of about less than one day. It obtains the forward-scattering near-field represented by spherical harmonics function [19]. Field values within a certain solid angle are then accumulated to get responses on the QPD. Coupling has been revealed in OT, where the diameter of laser beam waist at focus and spherical particle was both about 1 μm [11]. However, it expresses far-field with endless distance approximation. Thus, an interference image at a limited distance cannot be acquired this way. We yield the interference image through twice Fresnel diffraction [20], as shown in Figure 2. The first diffraction is calculated with the single fast Fourier transform (SFFT) method and the second one with the double fast Fourier transform (DFFT) method.

#### 3.2.2. Computation Complexity

For the OT structure we used, the derivation below shows that 253 instances of computation are needed if DFFT is chosen as the first Fresnel diffraction method compared with the SFFT. Furthermore, 4 instances of computation are required for the SFFT compared with the DFFT in the second diffraction. Figure 3 shows the sketch of in the first diffraction process with the SFFT method and the DFFT method is similar with it. The lateral size of the main energy region (MER) near the particle is $L_{1}$ and is $L_{2}$ for far-field on the front surface of condenser. $L_{2}\approx 2f_{1}\tan(\mathrm{asin}\left(\mathrm{NA}/n_{\mathrm{med}}\right)$. The number of sampling points both at the near and far field is $N_{s}$. The grid size of diffraction surface is $L_{3}$ while that of the observation surface is $L_{4}$. $L_{4}=N_{s}\lambda L_{5}/L_{3}$ in the SFFT method where $L_{5}$ is the diffraction distance and $L_{5}=f_{1}$ in the first diffraction process. $L_{4}=L_{3}$ in the DFFT method. $N_{s}$ is the total number of sampling points. Therefore, two sets of conditions need to be met. First, the sampling space needs to cover MSR. Second, the number of sampling points in MSR is not less than $N_{min}$. Those are:(6)$$\left\{ \begin{array}{l} L_{3}\geq L_{1}\\ L_{4}\geq L_{2}\;\\ N_{s}L_{1}/L_{3}\geq N_{\min}\\ N_{s}L_{2}/L_{4}\geq N_{\min} \end{array}\right..$$
The values for $N_{s}$ are at least 439 and 100 in the first and second diffraction, respectively, when $N_{\min}$ is equal to 100. For the single FFT (SFFT) method, the grid size of diffraction surface is inversely proportional to the observation surface. For the double FFT (DFFT) method, the grid sizes of them are equal. While the grid number always keeps the same during the diffraction in SFFT and DFFT. Thus, SFFT is more suitable for diverging or converging diffraction process while DFFT for approximately parallel beams because of much less computation for the same accuracy.

#### 3.2.3. Computation Errors

The SFFT and DFFT methods are both based on scalar diffraction. The intensity distribution of the image rather than the phase is concerned there. Paraxial approximations should therefore be taken into account [21], that is:(7)$$L_{5}\gg L_{6}=\sqrt{2{\left(L_{2}/2\right)}^{2}}/2.$$
$L_{5}$ is 18.4 mm and $L_{6}$ equals to 2 mm in the first diffraction process. $L_{5}$ is 200 mm and $L_{6}$ equals to 2 mm in the second diffraction process. The grid size of diffraction surface also needs to be much larger than wavelength, hence:(8)$$L_{3}\gg \lambda_{0}.$$
$L_{3}$ is 80 μm and 11.2 mm in the first and second diffraction process, respectively. Moreover, $\lambda_{0}$ equals to 1.064 μm

Only the forward scattered field is considered above, while the backward field of the second beam is ignored in OT with counter-propagating beams. Simulation shows that backward field intensity is two orders of magnitude lower than that of the forward one because of the low refractive index and Mie scattering state.

## 4. Simulations

### 4.1. Simulation of Computing Forward Scattered Far Field

For the OT structure we used, the images in Figure 4a show the simulation results for the far-field interference image when the particle moves the distance $x_{x}$ along the x-axis. The images in Figure 4b are simulation results for the scenario when particle moves the distance $x_{z}$ along the z-axis. The particle is on the optical axis and is 10 μm away from the focus of Beam No.1 in Figure 1, along the beam propagation direction when $x_{x}=x_{y}=0$ and $x_{z}=0$. More specifically, the particle is closer to the condenser compared with the focus of Beam No.1. Curves in Figure 4c are the normalized laser intensity distributions of the interference image in horizontal cross-section in Figure 4a, whereas curves in Figure 4d are those of Figure 4b by the same token.

It can be seen from Figure 4a,c that the interference image is similar to an Airy disk when the particle is exactly on the optical axis. These images all have a bright spot and ring with an outer diameter of about 8 mm. Conversely, only the spot is present if particle size is much smaller than beam waist as Figure 4e shows, which can be explained qualitatively by the diffraction principle. The spot is shifted to left side when the particle moves right, but the ring stays in place. It is clear from Figure 4c that the ring intensity increases on the left and decreases on the right, while spot intensity is almost fixed. Based on Figure 4d, the intensities of the spot and the ring both gradually decrease when the particle moves along the beam propagation direction. The former changes more obviously, while their central positions remain unchanged. Since the images in Figure 4a reflect $x_{x}$ directly, they can also instruct the alignment, as well as monitor structural changes, of OT.

### 4.2. Simulation of Decoupling with the Modified QPD Method

A difference in laser intensity between the two halves of the interference image corresponds to a radial signal in QPD. Neither the spot nor the ring in the image is divided equally by QPD when the particle is not on the optical axis as in Figure 4a. Therefore, the radial detector responds to axial motion in that case. If an iris is set before QPD to filter out the ring and retain the spot in the image, the equally divided spot provides information of lateral motion only. The axial signal is derived by the total laser intensity variation of the image. When the particle is not on the optical axis, axial coupling-detection occurs, since laser intensity changes on both sides of the ring cannot cancel each other out, as seen in Figure 4c. The setting of an iris eliminates the ring and is expected to suppress this coupling.

Assuming that the two beams only have nonzero $x_{x}$ in the x-axis direction in the OT structure we used, Figure 5a shows the relationship between $x_{x}$ and $R_{\mathrm{xz}}$ when using the conventional QPD method and the modified QPD method in interference image simulation, with different iris diameters $D_{\mathrm{iris}}$. Similarly, the relationship between $x_{x}$ and $R_{\mathrm{zx}}$ is described in Figure 5b. $R_{\mathrm{xy}}=$
$R_{\mathrm{yx}}$ and their value is always no more than −80 dB even if the $x_{x}$ deteriorates to 1 μm, thus no further analysis for them is provided below.

The coupling-detection ratios $R_{\mathrm{xz}}$ and $R_{\mathrm{zx}}$ grow rapidly as $x_{x}$ increases. When $x_{x}$ reaches to 0.3 μm in conventional QPD, these will increase to −16.7 dB and 13.5 dB, respectively. When iris diameters in modified QPD are set to 3 mm and 5 mm, respectively, the values for coupling suppression are optimal radially and axially, reducing to −25.7 dB and −23.2 dB, respectively, when $x_{x}$ equals to 0.3 μm. That consists a diminution of coupling detection level up to 9.0 dB radially and 36.7 dB axially. As $x_{x}$ increases, the amplitude of coupling repression also attenuates rapidly. This means that the modified QPD method can have a better coupling suppression effect, if the alignment is improved in advance. Figure 5c takes on the relationship between $x_{x}$ and $\beta_{\mathrm{xx}}\;$, while Figure 5d demonstrates that $\beta_{\mathrm{zz}}$. $\beta_{\mathrm{xx}}$ decreases by about 6.6 dB, when using 3-mm iris as an optimal diameter. Furthermore, $\beta_{\mathrm{zz}}$ increases by an order of magnitude only in modified QPD. Figure 5e shows that the x-axis detection nonlinearity $\eta_{\mathrm{xx}}$ will not exceed −46.5 dB when $x_{x}\leq 1\;\mu m$ under all kinds of diameters of the iris before QPD and it rapidly decays with $x_{x}$. As for the z-axis detection, the nonlinearity is relatively worse and basically stays between −10 dB to −60 dB within 4 μm of z-axis motion in Figure 5f.

## 5. Experiments

### 5.1. Experiment of Recording Forward Scattered Far Field

The experiments were performed as follows: First, a piezoelectric ceramics excited by high voltage signal was used to vibrate a glass substrate. The sphere originally stuck on the glass substrate fall into the laser focus and was trapped by the laser. Second, we recorded the triaxial QPD signals under two conditions. The first condition is that there are no irises before the triaxial QPD photosensitive surface, and the second one is that the diameters of the irises were adjusted carefully to optimize coupling suppression effect as good as possible. Lastly, the electronic signals of QPD output were collected by a signal acquisition card (Model PCI-4472, National Instruments, Inc., Austin, TX, USA) and transmitted into PC. The PC computed the displacement power spectrum with the time domain data in real time for the convenience of adjusting irises. The laser beam profiler (Model WinCamD-LCM4, DataRay, Inc., Redding, CA, USA) and the adjustable irises (Model SM1D12CZ, Thorlabs, Inc.) both came from USA.

The far-field interference image was recorded by a beam profiler as in Figure 6a, with its vertical cross-section shown in Figure 6b. In the latter figure, the cross-section (blue solid line) is compared with the simulation curve (red dotted line), when $x_{x}=0$.-When considering the relative central position of the spot and the ring in Figure 6a, $x_{x}$ is no more than 0.3 μm. In the experiment, $x_{x}$ was mainly concentrated in the horizontal direction and the spot was slightly down in the vertical direction. First, a pinhole of 5-μm-diameter was put into OT and we aimed the focuses of two beams at it. However, the thickness of the pinhole cannot be infinitely small, with the one we selected of about 50 μm. It is difficult to guarantee a coaxiality of less than 1/25 radian and ensure at most 1 μm $x_{x}$ in free space by manual adjustment. The particle will be at the midpoint of the line connecting the focal points, assuming that the counter-propagating dual-beams are exactly the same. We used images in Figure 4a to instruct alignment, and then Figure 6a was acquired.

### 5.2. Experiment of Decoupling with the Modified QPD Method

A variable iris was placed in front of QPD on each axis in modified QPD method in experiments, as close to it as possible to reduce the diffraction effect. The simulations in Figure 5a,b indicate that the iris should eliminate the ring in the radial detection entirely and keep half of that axially.

In theory, coupling-detection ratio between axes $R_{ij},\;i\neq j$ can be measured by exerting a known force on the levitated particle in OT. However, this is not easy to implement in practice. The thermal motion of the particle in OT will drown out the known force signals that we applied under relatively high gas pressure. On the other hand, increasing the known force exerted may cause the particle to deviate too far from the linear response region of the QPD detection, to accurately measure the coupling ratio.

In contrast, the displacement power spectrum (PSD) of the particle is a more convenient tool for checking the coupling ratio, as it is unnecessary to add devices of applying known forces into the OT and choosing the magnitude of the force carefully. Displacement PSD on the j-axis in OT can be described as [22]:(9)$$S_{xx,j}(\omega )=2<x_{j}^{2}>f_{L}(\omega ),$$
where $f_{L}\left(\omega \right)$ is the normalized Lorentz function: $\Omega_{j}^{2}\Gamma_{0}/\left[{\left(\Omega_{j}^{2}-\omega^{2}\right)}^{2}+\omega^{2}\Gamma_{0}^{2}\right]$, $\Gamma_{0}$ is gas damping. And $\Omega_{j}$ is intrinsic angular frequency, which is close to angular frequency at the resonance peak in PSD. Next, the output voltage PSD of the *i*-axis motion detector, according to Equation (2) and Equation (4), is determined as follows:(10)$$S_{vv,i}(\omega )=\sum_{j=x,y,z}\beta_{ij}S_{xx,j}(\omega ).$$
The voltage PSD of the single axis detector shows multiple peaks when coupling results from the difference in $\Omega_{j}$ and irrelevance of motion between the axes. In general, $\Omega_{z}$ is much less than $\Omega_{x}$. Thus, $\beta_{\mathrm{xx}}^{2}S_{xx,x}\left(\Omega_{x}\right)\gg \beta_{\mathrm{xz}}^{2}S_{xx,z}\left(\Omega_{x}\right)$. When coupling is serious enough that $\beta_{\mathrm{xz}}^{2}S_{xx,z}\left(\Omega_{z}\right)\gg \beta_{\mathrm{xx}}^{2}S_{xx,x}\left(\Omega_{z}\right)$, $R_{\mathrm{xz}}$ will be:(11)$$\frac{S_{vv,x}(\Omega_{z})}{S_{vv,x}(\Omega_{x})}\approx \frac{\beta_{\mathrm{xz}}^{2}S_{xx,z}(\Omega_{z})}{\beta_{\mathrm{xx}}^{2}S_{xx,x}(\Omega_{x})}=R_{\mathrm{xz}}.$$
On the contrary, only the upper limit of the coupling ratio can be obtained under mild coupling. The limit is:(12)$$R_{\mathrm{xz},\max}\approx \frac{\beta_{\mathrm{xx}}^{2}S_{xx,x}(\Omega_{z})}{\beta_{\mathrm{xz}}^{2}S_{xx,x}(\Omega_{x})}=\frac{\beta_{\mathrm{xx}}^{2}}{\beta_{\mathrm{xz}}^{2}Q_{x}^{2}},$$
where $Q_{j}=\Omega_{j}/\Gamma_{0}\gg 3\;and\;j=x,y,z$. The ratio $R_{\mathrm{zx}}$ follows the same derivation process as above, except that the equal sign needs to be changed into the less-than sign in Equation (12).

As a result, the reduction of gas pressure can lower the upper limit of PSD method for checking the coupling ratio. The reduction of gas pressure also wakens the collision thermal movement and improves the signal to noise ratio of the above exerting known force method. However, the reduction of gas pressure cannot go on all the time because the levitated particle is more likely to escape from the OT in lower gas pressure. In order to ensure the particle stably exist in OT under low pressure, and the mechanical energy of it is continuously reduced, we need to apply appropriate feedback force on the particle according to the position and velocity information of it. This is the so-called cooling.

Coupling in detection may degrade the feedback cooling performance since the motions between axes are all random and uncorrelated to each other. Many application in ultra-weak mechanics sensing and quantum physics studies with OT also require cooling the particle motion in vacuum [11,12]. The possible relation between detection decoupling and cooling has not been analyzed explicitly and verified in experiment. However, the heating phenomenon possibly caused by feedback force coupling has been reported [12]. Both feedback coupling and coupling-detection lead to exert unwanted feedback force related with the motion of other axes on a specific axis. Therefore, we speculated that coupling-detection affects feedback-driven cooling to a certain extent.

Figure 7a shows $S_{vv,x}\left(\omega \right)$ in logarithmic form with conventional QPD or the modified QPD method with an optimal iris diameter. That is, $S_{vv,x}\left(\omega \right)\;\left(\mathrm{dB}\right)=10\log_{10}\left[S_{vv,x}\left(\omega \right)\right]$. Moreover, Figure 7c,e shows that of $S_{vv,y}\left(\omega \right)\;\left(\mathrm{dB}\right)$ and $S_{vv,z}\left(\omega \right)\;\left(\mathrm{dB}\right)$. The coupling ratio $R_{ij},\;i\neq j$ is then exactly the height difference between peaks at frequency $\Omega_{z}$ and $\Omega_{x}$ or $\Omega_{y}$ along the same curve in Figure 7a,c,e. Figure 7b,d,f shows all three axes displacements measured by QPD. The resonant peaks of three axes are quiet close to each other, relative to the measurement bandwidth. Therefore, it is not easy to observe the effect of decoupling in the time domain plot.

When using modified QPD instead of conventional QPD, $R_{\mathrm{xz}}$ was meliorated from 36.0 dB to −13.1 dB. This method improved $R_{\mathrm{yz}}$ from 9.5 dB to −10.1 dB and also reduced $R_{\mathrm{zx}}$ from −5.5 dB to −28.4 dB. That is, a reduction of coupling-detection ratio occurs up to 49.1 dB radially and 22.9 dB axially. Signal strength increases by 14.3 dB at the x-axis and 3.4 dB at the y-axis around the resonance peak and decreases by 11.0 dB radially in the range of 500 Hz to 10 kHz. According to Figure 7a,c,e, there is no loss of bandwidth.

In generally, for particles whose sizes are relatively smaller, ranging from tens of nanometer to hundreds of nanometer, the resonance peaks in displacement power spectrum of all three axes are located at tens of kilohertz or even hundreds of kilohertz. Such high resonance frequencies make the main motion components of three axis of small-sized particle are separated in the frequency domain. In that case, coupling can be reduced to a low level by narrowband filter. For example, three different lock-in amplifiers were used to extract triaxial motions of particles of diameter ranging from 26 to 160 nm in reference [23]. However, as Figure 7 shows, the main motion components are basically coincident in the frequency domain for large-sized particle, making the narrowband filter ineffective.

## 6. Discussion

Apart from misalignment, there are other factors that affect detection accuracy. When using conventional QPD methods, $R_{\mathrm{xz}}\;$= 36.0 dB for the red line in Figure 6a, that is much higher than that in the simulation, as in Figure 4a. This appears to be caused by asymmetric beam shape, increasing off-centering of the spot division in the interference image. The laser intensity on the right side of the ring in Figure 5a confirms this, showing that modified QPD can suppress it and spatial filters may help to improve beam quality.

The voltages PSD of the red line in Figure 6a and the blue line in (b) take on apparent second and third harmonics of the pretty large amplitude of z-axis motion. It indicates a nonlinear relation between $V_{jz}\;\left(j=x,y,z\right)$ and $x_{z}$. Nonlinearity is less than −25.1 dB for the second harmonic and −32.5 dB for the third one, which is consistent with the simulation results in Figure 5f. Furthermore, it can be reduced by of other means of detection or by controlling the z-axis motion of the particle.

## 7. Conclusions

Overall, optical tweezers with counter-propagating beams and a large-sized particle is a technology with great potential, which is applicable in fields of research such as cell biology, weak mechanics sensing and quantum physics. However, its measurement accuracy faces challenges due to coupling of different axes. This paper proves that coupling is caused by misalignment from the perspective of the forward-scattering far-field.

A new method of suppressing coupling is proposed herein, adding a variable iris in front of the QPD photodetector on each axis. Experiments show a reduction of 49.1 dB in coupling-detection ratio radially and that of 22.9 dB axially, which is basically in accord with simulations. These make it possible to accurately measure three-dimensional mechanical quantities with OT synchronously and may be helpful to actively cooling the particle even to quantum ground state in ultra-low gas pressure in basic physics research. The measurements of bandwidth and signal strength do not suffer a great loss in these methods as opposed to those such as increasing beam diameter. A laser beam profiler and three variable irises are added into OT without changing the original optical structures of levitation and detection. Thus, they provide simple operation at a low cost and structural compatibility.

## Figures and Tables

**Figure 1 sensors-20-04916-f001:**
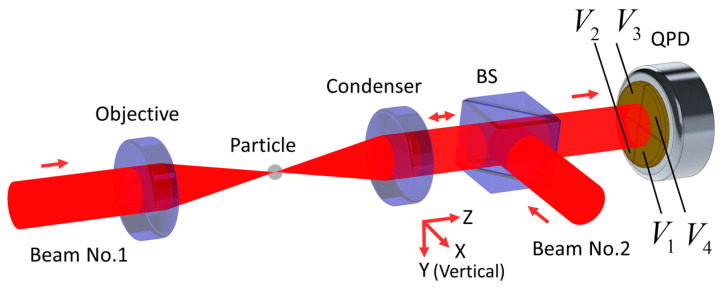
Typical as four-quadrant photodetectors (QPD) scheme of particle centroid displacement detection in optical tweezers (OT). Beam No. 1 is incident on the particle after being focused by the objective, whose focus approximately coincides with the condenser, thus incident light and forward scattered light are both collected by the condenser. Once they have passed through the beam splitter (BS), a far-field interference image is formed on the photosensitive surface (yellow) of the QPD. In the following simulation and experiment discussions, the direction of gravity is always along the positive y-axis.

**Figure 2 sensors-20-04916-f002:**
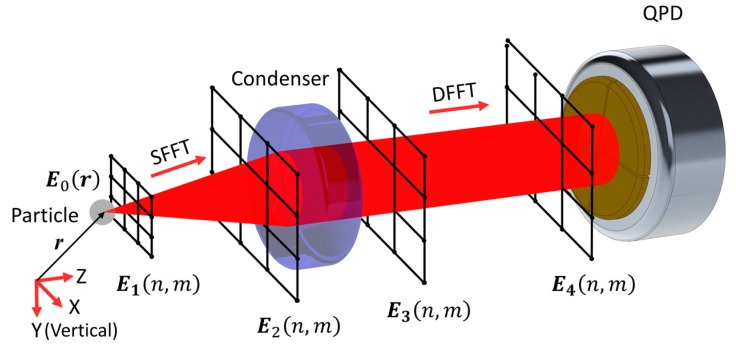
Schematic of obtaining the far-field interference image by twice Fresnel diffraction. First, near-field $E_{0}\left(r\right)$ is changed into sample values $E_{1}\left(n,m\right)$ in a Cartesian coordinate system. Second, field $E_{2}\left(n,m\right)$ on the front surface of condenser is calculated by the single fast Fourier transform (SFFT) method. It is Then multiplied by the lens phase function and turns into $E_{3}\left(n,m\right)$. Finally, the interference image $E_{4}\left(n,m\right)$ on the surface of the QPD is obtained by the double fast Fourier transform (DFFT) method.

**Figure 3 sensors-20-04916-f003:**
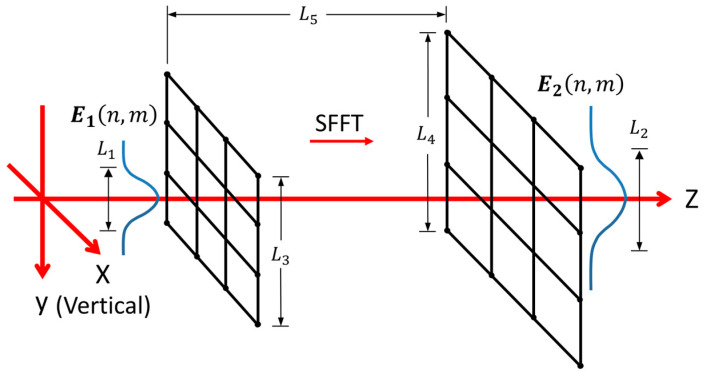
Sketch of the first diffraction process with the SFFT method. The parameters in the first diffraction process with the DFFT method are marked with the same symbols.

**Figure 4 sensors-20-04916-f004:**
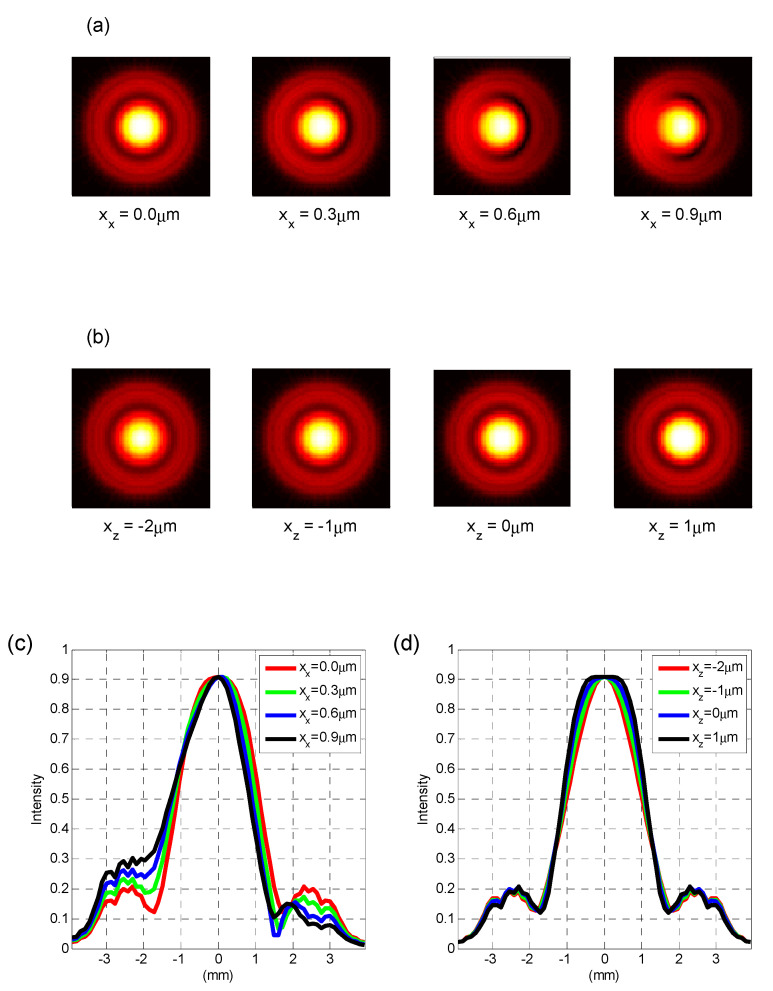
(**a**) Far-field interference images of a 10-μm-diameter sphere in simulation under different radial displacements. Other parameters about the sphere and beams were explained in the above experiments and simulation setup; (**b**) far-field interference images of a 10-μm-diameter sphere under different axial displacements; (**c**) horizontal cross-section of (a); (**d**) horizontal cross-section of (b); (**e**) far-field interference images of a 10-nm-diameter sphere in simulation when $x_{x}=0$ and $x_{z}=0$. The polystyrene sphere is levitated by a NA = 0.9 single laser beam in water. Thus, the beam waist diameter at focus is about 1.33 μm. A NA of 1.0 condenser is 1.5 mm away from the sphere and collects the scattered light. The distance between sphere and beam focus is about 0.1 μm; (**f**) horizontal cross-section of (e). (see Appendix A).

**Figure 5 sensors-20-04916-f005:**
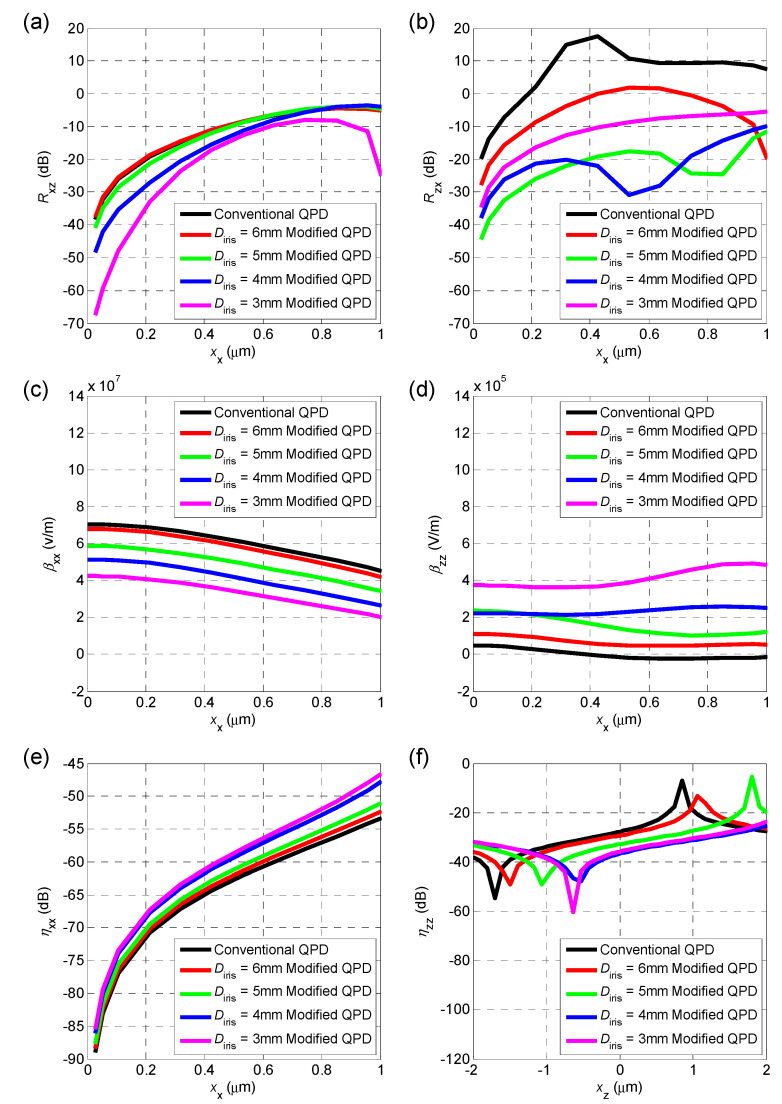
(**a**) X-axis coupling-detection ratio resulting from the z-axis motion $R_{\mathrm{xz}}$ under different $x_{x}$ in conventional QPD and modified QPD methods; (**b**) relationship between the z-axis coupling-detection ratio caused by x-axis motion $R_{\mathrm{zx}}\;$ and $x_{x}$ in conventional QPD and modified QPD methods; (**c**) x-axis motion detection sensitivity of x-axis detector $\beta_{\mathrm{xx}}$ under different $x_{x}$ values; (**d**) relationship between the sensitivity of z-axis detector $\beta_{\mathrm{zz}}$ and $x_{x}$. (**e**) x-axis detection nonlinearity $\eta_{\mathrm{xx}}$ under different x displacement $x_{x}$. (**f**) z-axis detection nonlinearity $\eta_{\mathrm{zz}}$ under different z displacement $x_{z}$. (see Appendix A).

**Figure 6 sensors-20-04916-f006:**
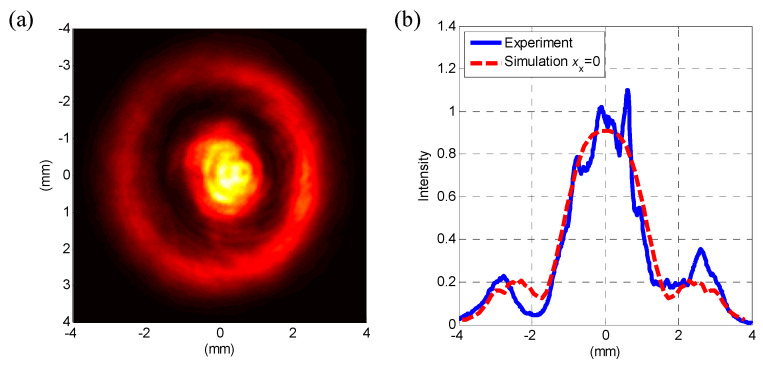
(**a**) Interference image before QPD in the experiment; (**b**) vertical cross-section of (**a**) with the simulation curve.

**Figure 7 sensors-20-04916-f007:**
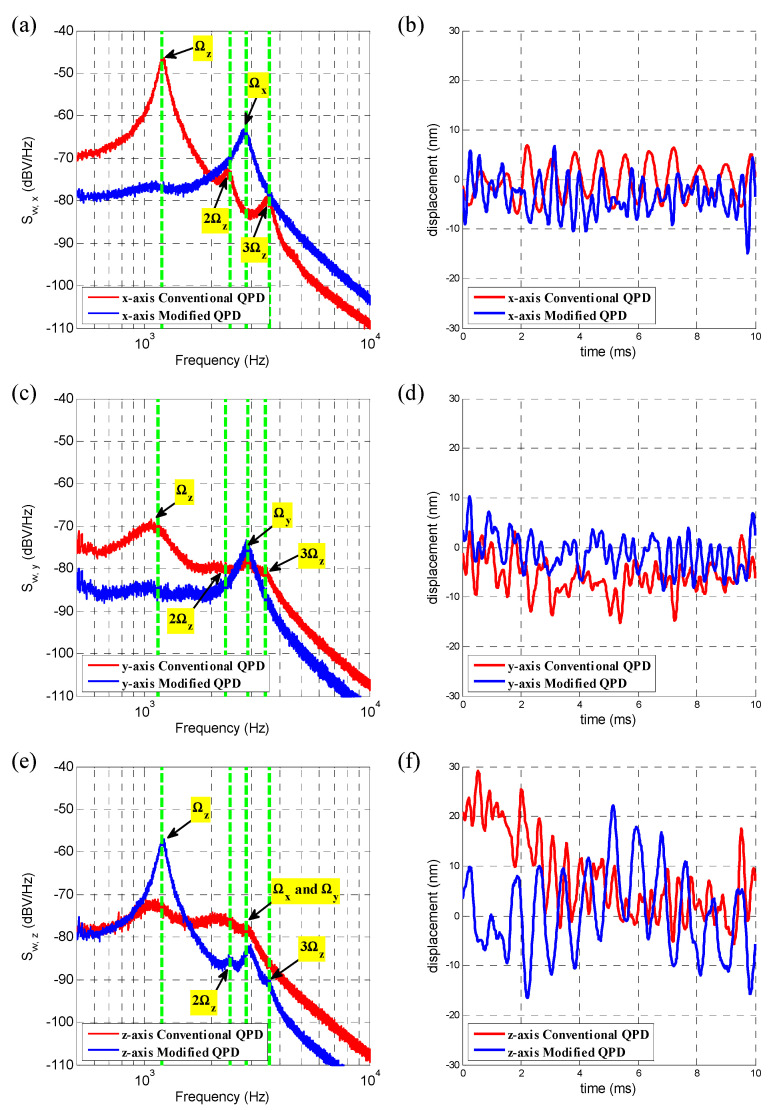
(**a**) Output voltage displacement power spectrum (PSD) of x-axis motion detector $S_{\mathrm{vv},x}\left(\omega \right)$ (dB) in conventional QPD and modified QPD method in experiment. Green dash lines added for comparing PSD curves at resonant peaks; (**b**) x-axis displacement of the sphere measured by the x-axis detector in conventional QPD and modified QPD method in experiment; (**c**) $S_{\mathrm{vv},y}\left(\omega \right)$ (dB) in conventional QPD and modified QPD method; (**d**) y-axis displacement in conventional QPD and modified QPD method; (**e**) $S_{\mathrm{vv},z}\left(\omega \right)$ (dB) in conventional QPD and modified QPD method; (**f**) z-axis displacement in conventional QPD and modified QPD method.

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
