# Peer review of "Displacement Detection Decoupling in Counter-Propagating Dual-Beams Optical Tweezers with Large-Sized Particle"

_sensors, 2020, doi:10.3390/s20174916_

Round 1

Reviewer 1 Report

Zhu et al. have presented a method to reduce noise signal coupling between different axis in a counter-propagating dual-beam optical tweezers setup by using irises. Although I assume many labs are using the iris method for noise reduction routinely for noise coupling reduction, I am not aware of any paper giving a detailed and focused discussion on this aspect. I thus think the current paper will provide a useful guidance for researchers in related fields. The overall quality of the paper is good, with conclusions well supported by evidences. I would thus recommend the acceptance of the paper after minor revisions.

  1. I think there are some typos in page 9 and 10, where sometimes "QPD" is referred to as "CPD" instead. I think all the "CPD"s are typos in the paper. If not, the authors should clearly define what "CPD" refers to.
  2. In Figure 6, it is clear that adding an iris is lowering the PSD at certain frequency range, while increasing it in other frequency ranges. I find the current labeling scheme for resonance peaks (Ωz or Ωx) are quite misleading. It almost seems that these resonances do not exist in the other curve, which is not the case. I would thus recommend using dash lines to indicate the positions of the resonance peaks.

Reviewer 2 Report

The paper presents a simple, yet effective, technique to improve sensitivity and reliability of position measurements in optical trapping of large dielectric particles, up to ten micrometers in diameter. Experimental and simulated results, the latter obtained via numerical computations of the Fresnel diffracted field, are discussed.

The idea behind the technique is quite a basic one: adjustable irises are placed in front of the four-quadrant detectors used to monitor the particle position owing to displacement of the far-field interference. I cannot judge whether the technique is truly innovative, or not. Cutting diffracted beams is a rather customary operation in optical experiments, but I think the rather detailed analysis presented in the paper is worth of publication. Moreover, the text is sufficiently clear and well organized, and results are generally convincing, in my opinion.

Prior to acceptance, however, the Authors are suggested to consider the following points.

  1. It is clear that the technique aims at improving displacement detection in optical tweezer experiments, in particular by suppressing crosstalk between different Cartesian directions, rather than directly enhancing the optical trapping efficiency. At lines 22-23 (Abstract) possibilities are mentioned relating with active cooling: the reader might understand that the paper aims at directly enhancing cooling capabilities. In my opinion, the outcomes of the proposed technique in terms of feedback-driven cooling are to be demonstrated, and this should be clearly pointed in the Abstract.
  2. A similar comment applies also to the paragraph at lines 45-57 in the Introduction. If I well understand, the proposed technique can be useful to reduce misalignment owing to the improved sensitivity in one-directional measurements, but it does not directly affect the extent of radial alignment errors in the trapping beams. Also in this case, Authors should rewrite the relevant sentences in order to better anticipate results discussed in the text.
  3. Discussion of the results of reference [10] at lines 58-60 (Introduction) is unclear: Authors are suggested to go more deeply through the content of that reference and account for that in a more clear way (my opinion is that the quoted work does not imply any well-defined limit in the particle diameter for a displacement measurement to be carried out).
  4. The Authors should explain why four-quadrant detector signals have to be time-averaged, as mentioned at line 99: is it a functional (instrumental) or measurement-related requirement?
  5. The meaning of “zero frequency area” (line 111) is obscure to me and should be clarified.
  6. The Authors should better assess linearity (line 117) of the displacement measurements enabled by four-quadrant detectors in practical configurations. This is a rather crucial point, in my opinion, for the subsequent discussion and I think that the reader might find very useful a quantitative evaluation of the non-linear contribution to the measurements.
  7. How can the Authors state that the particle is “10 um away from the focus” (lines 128-129)? Can such a distance be measured?
  8. Use of single and double-FFT in the calculations (lines 145 and following) should be better explained. Further to computational requirements, which are sufficiently clear in the present version, are there physical reasons for the adopted choice?
  9. The physical meaning of Eq. 6 should be highlighted, or a relevant reference given. At present, the mentioned condition is rather obscure to me. More in general, whenever possible it would be useful to have a graphical description (a figure) of the relevant distances.
  10. The statement at lines 197-198, on the occurrence of the spot only when the particle is smaller than the beam waist, is unclear (is it correct?) and would take advantage of a dedicated plot.
  11. I could not find the size of the sensitive area for the detectors used in the experiment. It would be a useful information.
  12. The statement at lines 288-289 would require discussions or a relevant reference (see also my comment 1 above).
  13. The acronym CPD used in several instances is not defined (and presumably it should read QPD): Authors must check and fix the issue.

Reviewer 3 Report

The authors declare that this study analyzes the source of coupling in a forward scattering far-field detection regime and then proposes a novel method of suppression for detecting the displacement of a large-sized particle with double-beam optical tweezers (OT). My suggestions for this manuscript are in the following.

  1. No clear explanation (or citation) for the statement: The minimum measurable acceleration in OT is inversely proportional to the square of the diameter of the spherical levitated particle.
  2. The title mentions “displacement detection”, however, the authors didn’t present the experimental data of particle displacement measurement, even no simulation result as well. It is difficult to evaluate the value of this paper.
  3. The direction of gravity is not marked in Fig. 1, which is crucial for measuring a levitated 10 micro-meter sphere particle.
  4. Why did the authors use low NA (0.15) in this study? Why not high NA? In low NA situations, the scattering force of OT is much larger than the gradient force. In the case of trapping large-sized particles, I doubt the low NA can stably trap the particle. Especially, if the gravity direction is along Y-axis, the trapping stability will be unlikely.
  5. In section 2 (Materials and Methods), the experimental setup is not very clear. The authors should present the providers of the instruments (i.e. objective, condenser, QPD, laser sources) and the sphere particles. Moreover, the authors should explain how the experiments were performed, i.e. the operation steps of the particle trapping and the record of the QPD signals.

Round 2

Reviewer 2 Report

The Authors demonstrated to have taken into account all criticisms raised in my original review. They devoted sufficient efforts to prepare a revised version of the manuscript addressing those criticisms.

Although the presentation style still suffers from some language issues, likely to be fixed at the editorial stage, the manuscript is now improved in terms of clarity, completeness, and overall scientific soundness. In particular, motivations and goals are now more sharply provided to the interested reader, who can find the paper stimulating in order to conceive more accurate experiments involving also feedback and active cooling mechanisms.

Therefore, in my opinion the paper can be accepted for publication in the present form, provided that language revisions are implemented at the editorial stage.

Author Response

Thank you very much for your suggestion. We will ask for the specialist language editing service and solve the language issues.

Reviewer 3 Report

I have reviewed the revised manuscript again. On the whole, the quality of the revised manuscript has been improved. I still have two questions in the below.

  1. According to the authors’ explanation, the minimum measurable acceleration in OT should be inversely proportional to the “square root” of the diameter of the spherical levitated particle, but not “square”. I suggest the authors to carefully check the proposed formula derivations.
  2. I am confused by that the authors just measure/simulate x- and z-axes displacements, but exclude y-axis. I suppose that y-axis displacement should be much important than z-axis. Could the authors explain why they chose the x- and z-axes to study?
